# Structurally and Morphologically Distinct Pathological Tau Assemblies Differentially Affect GVB Accumulation

**DOI:** 10.3390/ijms241310865

**Published:** 2023-06-29

**Authors:** Marta Jorge-Oliva, Jan R. T. van Weering, Wiep Scheper

**Affiliations:** 1Department of Functional Genomics, Center for Neurogenomics and Cognitive Research, Vrije Universiteit Amsterdam, Amsterdam Neuroscience—Neurodegeneration, 1081 HV Amsterdam, The Netherlands; m.jorgeoliva@vu.nl (M.J.-O.); j.vanweering@amsterdamumc.nl (J.R.T.v.W.); 2Department of Human Genetics, Amsterdam UMC Location Vrije Universiteit, Amsterdam Neuroscience—Neurodegeneration, 1081 HZ Amsterdam, The Netherlands

**Keywords:** Tau, aggregation, granulovacuolar degeneration bodies, neurodegenerative disease

## Abstract

Tau aggregation is central to the pathogenesis of a large group of neurodegenerative diseases termed tauopathies, but it is still unclear in which way neurons respond to tau pathology and how tau accumulation leads to neurodegeneration. A striking neuron-specific response to tau pathology is presented by granulovacuolar degeneration bodies (GVBs), lysosomal structures that accumulate specific cargo in a dense core. Here we employed different tau aggregation models in primary neurons to investigate which properties of pathological tau assemblies affect GVB accumulation using a combination of confocal microscopy, transmission electron microscopy, and quantitative automated high-content microscopy. Employing GFP-tagged and untagged tau variants that spontaneously form intraneuronal aggregates, we induced pathological tau assemblies with a distinct subcellular localization, morphology, and ultrastructure depending on the presence or absence of the GFP tag. The quantification of the GVB load in the different models showed that an increased GVB accumulation is associated with the untagged tau aggregation model, characterized by shorter and more randomly distributed tau filaments in the neuronal soma. Our data indicate that tau aggregate structure and/or subcellular localization may be key determinants of GVB accumulation.

## 1. Introduction

Tauopathies comprise a subset of neurodegenerative diseases that include Alzheimer’s disease (AD) and frontotemporal dementia (FTD) [1]. Tau pathology can be influenced by genetic, epigenetic, and environmental factors [2]. Mutations in the tau-encoding gene *MAPT* give rise to familial FTD in an autosomal dominant manner [3], demonstrating the causal relationship between tau pathology and neurodegeneration. In addition, tau pathology in AD is the closest correlate to neurodegeneration in post-mortem studies [4], which was more recently confirmed by tau positron emission tomography (PET) imaging [5]. In line with this, tau is one of the major therapeutic targets investigated for AD and FTD [6,7]. Despite the key role of tau aggregation in this large group of diseases, it is still unclear how tau pathology causes neurodegeneration and the molecular mechanisms that brain cells employ to prevent it.

The accumulation of aggregated tau proteins indicates a major disturbance of protein homeostasis (“proteostasis”), a delicate balance between the synthesis and degradation of proteins. Proteostasis is essential for cell viability and is therefore tightly controlled by proteostatic stress responses that are activated to counter challenges to proteostasis via effects on protein synthesis, folding, and degradation [8,9,10,11]. The neuron-specific aspects of proteostatic stress resilience and vulnerability are only now beginning to emerge and may provide important targets for therapeutic intervention.

A striking neuron-specific response to tau pathology is presented by granulovacuolar degeneration bodies (GVBs). GVBs are proteolytically active lysosomal organelles carrying a dense core that is strongly positive for markers of cellular stress responses [12,13], indicating a connection with the response to disturbed proteostasis. Previous work from our lab demonstrated that in the brains of AD patients, GVBs are predominantly found in neurons with early stages of tau aggregation and not in neurons that have end-stage pathology [14,15], suggesting that GVBs form as an early neuronal stress response to pathological tau accumulation. Interestingly, cognitively healthy centenarians have a relatively high GVB load compared to their tau load [16], which may suggest that augmented GVB accumulation may contribute to neuronal resilience to tau pathology.

The close connection of intraneuronal tau pathology to GVB accumulation may be relevant for understanding this potentially protective proteostatic response and therefore warrants further investigation. This is greatly facilitated by the development of cellular models for GVBs. Our lab reported the first primary neuron model for GVBs using viral transduction of human tau with the P301L FTD mutation fused to a green fluorescent protein (GFP) tag (FTDtau^1^-GFP) that forms intraneuronal aggregates upon the addition of exogenous tau pre-formed fibrils (PFFs) or seeds [17]. An inherent limitation of seeded aggregation models is the requirement for the addition of tau PFFs to elicit aggregation, which are intrinsically heterogeneous. Moreover, PFFs can introduce cellular phenotypes that are not directly related to intracellular aggregation via the direct interaction of the PFFs with the lipids and proteins in the plasma membrane. In particular, extracellular PFFs were shown to induce endolysosomal damage [18,19,20,21,22], which is a confounding factor in the study of GVBs. To circumvent this, we recently implemented a tau variant that combines FTDtau^1^ with an additional FTD mutation (P301L+S320F; FTDtau^1+2^) that has been shown to induce spontaneous pathological tau accumulation in mice in vivo [23] and ex vivo [24] to develop a novel GVB model in mouse primary neurons [25]. Similar to GVBs observed in the brains of tauopathy patients, GVBs in this spontaneously aggregating tau model present as vesicular structures with a membrane positive for lysosomal integral membrane protein 2 that contain a dense core immunopositive for casein kinase 1 δ (CK1δ) and other typical GVB markers [25].

Cellular aggregation models often make use of tau variants fused to fluorescent proteins such as GFP. The presence of a fluorescent tag facilitates direct detection of the protein of interest, thereby circumventing the need for immunodetection and allowing live cell imaging approaches to study aggregation. However, the presence of a GFP tag has been shown to affect the aggregation process of recombinant tau variants in vitro by steric hindrance [26]. Importantly, alterations in the aggregate structure are also likely to have an impact on downstream cellular responses tightly linked to tau pathological accumulation, like GVB formation. However, it is unknown how the presence of a GFP tag affects tau aggregation in the cellular context of the neuron. In the current study, we investigate the properties of tau aggregates and aggregation kinetics resulting from seed-independent untagged and GFP-tagged tau pathology and their differential effects on GVB accumulation.

## 2. Results

### 2.1. Pathological FTDtau^1+2^-GFP and FTDtau^1+2^ Assemblies Have Different Morphology and Subcellular Localization

To assess the pathological protein accumulations resulting from seed-independent aggregation of FTDtau^1+2^ either untagged or fused to a C-terminal GFP tag (FTDtau^1+2^-GFP), primary mouse neurons were lentivirally transduced with FTDtau^1+2^-GFP or FTDtau^1+2^. Cultures were transduced at day in vitro (DIV) 3 and fixed and analyzed after 15 days (Figure 1a). The characterization of the pathological protein accumulations resulting from the seed-independent aggregation of FTDtau^1+2^-GFP and FTDtau^1+2^ was done by immunolabeling with the MC1 antibody. Interestingly, the subcellular localization and morphology of FTDtau^1+2^-GFP and FTDtau^1+2^ MC1-positive tau assemblies were different (Figure 1b and Appendix A). Tau accumulation in the FTDtau^1+2^-GFP model showed clusters of pathological tau assemblies in the neuronal soma as well as in neurites, reminiscent of the observed aggregation of FTDtau^1^-GFP induced by K18 P301L tau PFFs [17]. In contrast, MC1-positive tau assemblies in the FTDtau^1+2^ model typically appeared more evenly distributed throughout the neuronal soma. Neuritic MC1-positive accumulations were less commonly observed in neurons in the FTDtau^1+2^ model. The total tau pathology load in these models was quantitatively assessed by single-cell analysis of the MC1 signal in neuronal somata. Tau pathology levels were not different between FTDtau^1+2^-GFP and FTDtau^1+2^ transduced neurons (Figure 1c). However, a trend for a decrease in the SD of MC1-positive tau accumulation in the FTDtau^1+2^ model was observed (Figure 1d), which is in line with the more evenly distributed MC1 signal observed in this model.

Methanol (MeOH) fixation to remove soluble tau was employed and showed that pathological tau accumulations in both the FTDtau^1+2^-GFP and FTDtau^1+2^ models are MeOH insoluble (Appendix A). FTDtau^1^-GFP was included as a non-aggregating control for MeOH treatment since it requires the addition of exogenous tau PFFs to aggregate in this experimental setup.

These data demonstrate that FTDtau^1+2^-GFP and FTDtau^1+2^ lentivirally transduced neurons form pathological insoluble tau assemblies that differ in morphology and subcellular distribution.

### 2.2. FTDtau^1+2^-GFP and FTDtau^1+2^ Display Similar Progression of Pathological Tau Accumulation

To study the implications of the observed differences between FTDtau^1+2^-GFP and FTDtau^1+2^ aggregates for the progression of pathological tau accumulation in neurons, we used high-content automated microscopy to quantify the intensity of the intraneuronal MC1 signal as a function of time (1–15 days) after lentiviral transduction (Figure 2a). This method confirmed our confocal data in Figure 1, showing that MC1-positive accumulation in the FTDtau^1+2^-GFP model was present in clusters in neuronal somata and neurites, whereas MC1 signal in the FTDtau^1+2^ model was predominantly found evenly distributed in neuronal somata (Figure 2b). Another difference was the number of neurons with MC1-positive inclusions, which was higher in the FTDtau^1+2^ model at any of the studied timepoints. After 15 days of exposure, 40–50% of neurons in the FTDtau^1+2^-GFP model contained pathological tau accumulations, compared to approximately 80% of the neurons in the FTDtau^1+2^-transduced condition (Figure 2b). No significant difference was found in the relative pathology increase (see Materials and Methods for analysis details) at subsequent time points between models (Figure 2c). Therefore, we conclude that FTDtau^1+2^-GFP and FTDtau^1+2^ have similar kinetics regarding the overall progression of pathological tau accumulation in neurons.

### 2.3. FTDtau^1+2^-GFP and FTDtau^1+2^ Result in Different Filament Ultrastructure in Neurons

Next, we assessed if the observed differences in tau aggregate morphology between FTDtau^1+2^-GFP and FTDtau^1+2^ models are associated with a difference in filament ultrastructure. Seeded aggregation of FTDtau^1^-GFP induced by recombinant K18 P301L tau PFFs was also included in this experiment to distinguish the effects of the presence of a C-terminal GFP tag from seeded and spontaneous aggregation.

Ultrastructural analyses revealed distinct filament characteristics and organization between the GFP-tagged and the FTDtau^1+2^ models (Figure 3a and Appendix A). Quantitative analysis of individual filaments demonstrated a significant difference in average length present in the ultrathin cross section (Figure 3b). Filaments in the GFP-tagged tau pathology samples showed similar average lengths of 322.2 nm (FTDtau^1^-GFP + PFFs) and 352.4 nm (FTDtau^1+2^-GFP), whereas filaments in the FTDtau^1+2^ model were much shorter (average 127.8 nm). No significant difference was found for the width of the filaments in the GFP-tagged models, which showed an average of 21.3 and 21.8 nm for the seeded and spontaneous FTDtau^1+2^-GFP models, respectively (Figure 3c). The average filament width in the FTDtau^1+2^ model was smaller, 16.6 nm, indicating that the C-terminal GFP tag generates wider filaments. Analysis of microtubule filaments, recognized by a hollow morphology in contrast to the solid tau filaments, showed no difference in filament width (Appendix A) between the different tau filament-containing neurons. This indicates that the observed differences in tau filaments are related to the tau variant and not to the neuronal context.

In addition, there is a clear difference in filament organization within the aggregate (Figure 3a). Filaments in both GFP-tagged models appear more organized in bundles of parallel filaments, whereas the filaments found in the FTDtau^1+2^ model are distributed in multiple orientations. To quantitatively assess the observed filament organization, we calculated the SD of the filament angle per analyzed aggregate in the field of view (Figure 3d). A higher filament angle SD shows more variation in the orientation of filaments within an aggregate cross section, which was the case for the FTDtau^1+2^ model. The filament angle SD was not significantly different between the FTDtau^1+2^-GFP and the seeded FTDtau^1^-GFP models. We conclude that filaments in the FTDtau^1+2^ model are thinner, shorter, and show a less organized bundle structure than those found in the seeded and spontaneous GFP-tagged tau pathology models.

### 2.4. FTDtau^1+2^ Is Associated with a Higher GVB Load Than FTDtau^1+2^-GFP

The presence of GVBs in the neuron is tightly associated with intracellular pathological protein accumulation. Our lab previously demonstrated that tau aggregation induces GVB formation in a model of recombinant K18 P301L tau seed-induced aggregation of FTDtau^1^-GFP [17] and more recently in an FTDtau^1+2^ unseeded aggregation model in primary mouse neurons [25]. Yet, it is still elusive which aspects of tau aggregation are involved in GVB accumulation. The spontaneous FTDtau^1+2^-GFP and FTDtau^1+2^ tau pathology models we characterized here result in differences in aggregate morphology and subcellular distribution (Figure 1) as well as ultrastructure (Figure 3). Since GVBs occur in both tau pathology models, they present a unique tool to further investigate the connection between tau aggregation and GVB formation.

We employed high-content microscopy to assess if these differences at the tau pathology level have an impact on the GVB load in these cultures. For this, we quantified the number of neurons containing GVBs using CK1δ immunopositivity in both the FTDtau^1+2^-GFP and the FTDtau^1+2^ models 15 days after tau transduction (standard protocol shown in Figure 1a), employing untransduced neurons as controls. The automated microscopy analysis showed that FTDtau^1+2^-GFP and FTDtau^1+2^ accumulate similar levels of MC1-positive pathological tau (Figure 4a,b). This is in sharp contrast with the difference in GVB load observed in the tau pathology models (Figure 4c,d). The percentage of neurons containing GVBs in the FTDtau^1+2^-GFP was 0.65%, whereas the GVB load was nearly 10-fold higher in the FTDtau^1+2^ model (5.07%). This large difference in GVB load observed in the different models suggests that tau aggregate structure and/or subcellular localization may be key determinants in GVB accumulation.

## 3. Discussion

Here, we studied the properties of different tau aggregates in relation to the accumulation of GVBs in neurons (see Figure 5 for a schematic overview of the results). Using a combination of confocal microscopy, TEM, and quantitative automated high content microscopy, we show that the presence of a C-terminal GFP-tag in spontaneously aggregating FTDtau^1+2^ strongly affects its aggregation characteristics in primary mouse neurons. Whereas both aggregation paradigms result in methanol-insoluble tau aggregates, FTDtau^1+2^-GFP assemblies appear as clusters in soma and neurites in contrast to the more evenly distributed and predominantly soma-localized FTDtau^1+2^ pathological accumulations. This is in line with the TEM analysis, where tau aggregates in the FTDtau^1+2^-GFP model are present as stacked large parallel filaments in contrast with the thinner, shorter, and less organized filaments observed in the FTDtau^1+2^ tau pathology model that intermix with organelles. The width of tau filaments resulting from our primary neuron tau pathology models was within the reported width range of filaments in the brains of FTDtau^1^ mice (10–30 nm) [27,28,29], which is also in line with observations in different human tauopathies [30,31]. The filaments in the seeded and spontaneous GFP-tagged models showed a similar width, which was significantly larger than that of FTDtau^1+2^ filaments. This suggests that the presence of a C-terminal GFP tag affects the width of the resulting tau filaments.

One of the advantages of FTDtau^1+2^ is that the model does not require exogenous PFFs, but the presence of two FTD mutations does not naturally occur in the human brain. However, the TEM ultrastructure of the filaments formed by FTDtau^1+2^ is highly similar to those observed in FTDtau^1^ mouse models [27,28,29] as well as in a human cell line overexpressing FTDtau^1^ treated with α-synuclein PFFs [32]. In addition, the aggregates of FTDtau^1^-GFP seeded with recombinant tau PFFs resemble the unseeded aggregates of FTDtau^1+2^-GFP, suggesting that the differences in aggregate properties we report here are mainly determined by the C-terminal GFP tag. By deduction, it would therefore be expected that tau assemblies of untagged FTDtau^1^ seeded with PFFs would resemble more the spontaneously forming FTDtau^1+2^ assemblies than PFF-seeded FTDtau^1^-GFP aggregates.

To induce tau aggregation, we used overexpression by viral transduction, which inherently introduces differences due to the use of different viral particles; therefore, a direct quantitative comparison of tau load should be interpreted with caution. Nevertheless, comparable MC1-positive tau accumulation is obtained in FTDtau^1+2^-GFP and FTDtau^1+2^ models 15 days after lentiviral transduction in our experiments. Importantly, our data also show that the presence of C-terminal GFP does not strongly affect the kinetics of MC1-positive tau accumulation. Despite reaching similar pathological tau levels, FTDtau^1+2^ results in a much higher GVB load compared to the tagged model. Although the number of neurons that accumulate pathological FTDtau^1+2^ assemblies is approximately 2-fold higher than that of FTDtau^1+2^-GFP, this would be insufficient to fully explain the order of magnitude difference in GVB load between models. Interestingly, FTDtau^1+2^ assemblies are predominantly localized in the soma, where the great majority of GVBs are detected. The differential subcellular localization of tau pathological species in the FTDtau^1+2^-GFP and FTDtau^1+2^ models could also contribute to the observed differences in GVB load. This could suggest that GVBs arise as well as accumulate in the soma rather than the neurites, but this requires further investigation.

GVBs are associated with early tau pathology in the human brain [14,15,33,34,35,36]. Previously, we demonstrated that neurons with GVBs in the human AD brain or in mouse primary neurons transduced with FTDtau^1+2^ always contain pathological tau accumulation [25]. Although there is an absolute requirement for the presence of pathological tau in the same neuron where GVBs are present, there is no direct quantitative correlation between the extent of pathology in an individual neuron and its GVB status [25]. This implies that GVBs can form in neurons with low levels of pathological tau assemblies, which is in line with GVBs being a response to early stages of tau pathology. Together with the data from the current study, this indicates that higher GVB accumulation is associated with specific properties of aggregates rather than just quantitatively more aggregation. It will be interesting to assess which properties these are and whether the process of GVB formation has already been initiated in response to the potential precursors of pathological tau assemblies like liquid–liquid phase separations [37]. Tau aggregation inhibitors may be employed in follow-up studies to provide further insight into the GVB-inducing properties of pathological tau assemblies.

Recently, cryo-EM studies have revealed that the structure of pathological tau assemblies is different within the spectrum of tauopathies (reviewed in [38]). Yet, GVBs form independently of the structure in the human tauopathy brain [13]. In addition, we have shown before that GVBs can also be induced by aggregates of α-synuclein [25]. Together, this indicates a role for GVBs as a more generalized response to pathological protein accumulation, independent of a specific primary sequence or secondary/tertiary structure. It will be interesting to investigate whether the presence of other pathological protein assemblies that co-occur with tau pathology in the brain—like beta amyloid in AD—affects the formation of GVBs. In the current study, we show differences between two aggregation models that have a strong effect on the downstream process of GVB formation. The models differ in the presence of a C-terminal GFP tag fused to mutant tau. Therefore, it is possible that the tag itself directly modulates GVB accumulation, for example by sequestering factors or vesicles involved in GVB formation. Our data demonstrate that the presence of this tag results in pathological tau assemblies with different properties. In view of the close connection between pathological protein assemblies and GVB formation, it is therefore likely that the GFP tag indirectly modulates GVB accumulation via its effect on tau aggregation. It is interesting to note that GVBs are much more prominently induced by the less densely packed and more disorganized FTDtau^1+2^ filaments. It could be speculated that GVB formation involves direct interaction with filaments that will be more accessible in the untagged tau aggregation model. Alternatively, this disorganized distribution may result in a local difference in proteostatic stress and, thereby, stress responses that may facilitate GVB accumulation. Until recently, GVBs were only known as a pathological phenomenon in the human brain, and their function is therefore still elusive. The observation that GVBs can form in neurons with very little pathological tau accumulation in the human brain as well as in experimental models [25] provides support for the position of GVB formation as an early proteostatic response to protein aggregation in neurons. Since GVBs are proteolytically active [17], they may help limit the protein accumulation itself or pathogenic pathways leading to neurodegeneration. The sequestration of tau kinases and cellular stress (reviewed in [12,13]) and proteins involved in apoptotic [39,40] and necroptotic [41] cell death pathways in the GVB core in the tauopathy brain could be a mechanism to prevent their action. Neuronal cell models for tau aggregation and GVB formation like the ones we describe here will be instrumental in further investigating the molecular mechanism and functional implications of GVB formation.

## 4. Materials and Methods

### 4.1. Primary Mouse Neuron Culture

WT C57BL/6 mice were employed to obtain primary neurons. Housing and breeding of animals were in accordance with institutional and Dutch government guidelines. All experiments were approved by the animal ethical committee of VU University/VU University Medical Center.

Animals were dissected at embryonic day 18.5. Cortices were isolated, and meninges were disposed of in an ice-cold Hanks’ balanced salt solution (Sigma-Aldrich, St. Louis, MO, USA) with 10 mM HEPES (Hanks-HEPES; Gibco, Billings, MT, USA). They were subsequently digested with 0.25% trypsin (Gibco) in Hanks-HEPES for 20 min at 37 °C. The tissue was 3× washed in Hanks-HEPES and triturated with fire-polished Pasteur pipettes in DMEM (Lonza, Basel, Switzerland) containing 10% heat-inactivated fetal bovine serum (Gibco), 1% penicillin-streptomycin (Gibco), and 1% non-essential amino acid solution (Gibco). Cells were centrifuged (5′ at 800 rpm) and gently resuspended in Neurobasal medium (Gibco) with 2% B-27 (Gibco), 18 mM HEPES, 0.25% Glutamax (Gibco), and 0.1% penicillin-streptomycin (NB+). Corresponding volumes from the resulting cell suspension were added per well to obtain 15,000, 40,000, or 300,000 cells/well densities in 96-, 24-, or 6-well plates, respectively. Plates and glass coverslips were previously coated with 5 µg/mL poly-L-ornithine (Sigma-Aldrich) and 2.5 μg/mL laminin (Sigma-Aldrich) overnight (O/N) at room temperature (RT).

Plated neurons were provided with 5% CO2 and kept at 37 °C. A volume of NB+ medium corresponding to 40% of the total well volume was supplied after 10 DIV to replenish nutrients and compensate for media evaporation.

### 4.2. Tau Pathology Induction

As previously described [25], spontaneous intraneuronal accumulation of pathological tau was achieved by lentiviral transduction of WT neurons with 2N4R human P301L, S320F tau in its untagged form (FTDtau^1+2^) or fused to a C-terminal GFP tag (FTDtau^1+2^-GFP) on DIV 3 in the case of the standard protocol and at the indicated time points in the case of Figure 2.

In some experiments, tau pathology was induced in neurons employing the seeded model described in [17]. Briefly, neurons were transduced with GFP-tagged 2N4R human P301L tau (FTDtau^1^-GFP) on DIV 3, and pre-sonicated K18 P301L tau PFFs were supplied to the medium to a final concentration of 100 nM on DIV 7. Tau PFFs were produced in-house using a protocol adapted from [42]. In brief, 40 µM recombinant Myc-tagged K18 P301L tau monomer (TebuBio, Le Perray, France) was incubated for 24h at 37 °C and 80 rpm with 20 µM low-molecular-weight heparin (Fisher scientific, Hampton, NH, USA) and 2 mM DL-Dithiothreitol (Sigma-Aldrich) in a 100 mM sodium acetate (Fisher scientific) solution at pH 7. The reaction mix was then centrifuged at 100,000× *g* for 30 min at 20 °C and the pellet was washed with an equal volume of 100 mM sodium acetate buffer (pH 7). The presence of cross-β-sheet structure and reaching the plateau of aggregation were monitored by Thioflavin T fluorescence. Fibrils were subsequently diluted to reach an approximate concentration of 10 µM and sonicated with a Hielscher UP200St ultrasonic homogenizer (two cycles of 1 min in pulses of 10 s and 5 s rest). The sonicated fibrils or seeds were aliquoted, snap frozen, and stored at −80 °C. Before use and after thawing, an additional round of sonication was performed (1 min in pulses of 10 s and 5 s rest).

Neurons either transduced with GFP-tagged 2N4R human P301L tau (FTDtau^1^-GFP) on DIV 3 or not subjected to lentiviral transduction were employed as controls. All constructs were cloned into lentiviral vectors under the cytomegalovirus promoter, and lentiviral particles were produced as described in [43].

### 4.3. Fixation of Primary Neurons

Neurons were fixed on DIV 18, employing a two-step protocol. Half of the culture medium was replaced by formaldehyde (FA; Electron Microscopy Sciences, Hatfield, PA, USA) in phosphate buffered saline (PBS; pH 7.4) to reach a concentration of 1.85% for 10 min at RT, followed by another 10 min at RT with 3.7% FA. In experiments where the removal of soluble proteins was required, MeOH fixation was used instead. Cultures were exposed to ice-cold MeOH at −20 °C for 15 min while gently shaking. After fixation, cells were washed with PBS (pH 7.4) and kept in solution at 4 °C.

### 4.4. Immunolabeling of Primary Neurons

Fixed neurons were permeabilized with 0.5% Triton X-100 (ThermoFisher Scientific, Waltham, MA, USA) in PBS (pH 7.4) for 5 min at RT and blocked for 30 min at RT with a blocking solution of 2% normal goat serum (Gibco) and 0.1% Triton X-100 in PBS. Incubation with CK1δ (Thermo-Scientific Cat# PA5-32129, 1:500), MAP2 (Abcam Cat# ab5392, 1:250), and MC1 (a kind gift from Dr. Peter Davies, 1:500) primary antibodies was performed O/N at 4 °C. Incubation with 1:500 Alexa Fluor (488, 546, and 647)-conjugated secondary antibodies (Invitrogen, Waltham, MA, USA) was performed for 1 h at RT. Antibodies were diluted to a working concentration in blocking solution. In some experiments, neurons were incubated with DAPI (Brunschwig Chemie, Basel, Switzerland) solution diluted in PBS (5 µg/mL) at RT for 5 min to visualize nuclei. Between and after antibody and DAPI incubations, cells were washed 3× for 5 min with PBS (pH 7.4).

Coverslips were mounted on microscope slides (ThermoFisher Scientific) with Mowiol (Sigma-Aldrich) and left to dry in the dark. Slides and PBS-filled 96-well plates were stored in the dark at 4 °C.

### 4.5. Confocal Microscopy and Image Analysis

Immunofluorescent imaging of slides was performed using a Nikon Eclipse Ti confocal microscope controlled by NisElements 4.30 software (Nikon, Tokyo, Japan) using a 40× oil immersion objective (NA = 1.3). Z-stacks with 5 steps (1 µm step size) were acquired. The laser settings were adjusted to prevent saturation in any of the channels.

Maximum-intensity projected Z-stacks were analyzed with ImageJ software (v1.53q; National Institutes of Health). Single-cell quantification of pathology in neuronal somata was achieved by measuring MC1 fluorescent intensity (mean and standard deviation (SD)) within a mask drawn based on the neuron-specific MAP2 signal. Normalized intensity values correspond to the ratio of individual values and the average value in the FTDtau^1+2^-GFP condition per experiment.

### 4.6. High-Content Microscopy and Automated Analysis

Primary neurons in 96-well plates were imaged on a CellInsight CX7 High-Content Screening (HCS) Platform (ThermoFisher) controlled by HCS Studio 2.0 Cell Analysis software (ThermoFisher) using a 20× air objective in widefield mode. Single focal planes were acquired except for the CK1δ channel, for which 3-step Z-stacks (3 μm step size) were acquired to include all GVBs. 40 randomly distributed fields were imaged per well, and the DAPI signal was used for automated focus.

Columbus analysis software (v2.5.2.124862; PerkinElmer, Waltham, MA, USA) and in-house developed scripts were used to analyze the resulting maximum intensity projections. The number of imaged neurons per well was obtained by quantifying neuronal nuclei, which were distinguished from non-neuronal nuclei by the morphology of the DAPI signal and its overlap with the neuron-specific marker MAP2. Nuclei in contact with the border of the field were excluded from the analysis. Neuronal tau pathology load was assessed by measuring the MC1 signal in a MAP2-defined mask. The GVB load was determined using the script employed in [25]. In brief, the protocol has a filtering step for the CK1δ signal to remove smooth and continuous background intensity. CK1δ-positive puncta in close proximity were defined as a cluster, and assessment of a cluster as a GVB-positive neuron was based on area, roundness, intensity, and overlap with the MAP2 signal. Measured parameters were corrected for the neuron number in the well and subtracted from the average (background) value measured in control wells per experiment. In the timeline experiment (Figure 2), corrected MC1 mean intensity was normalized by setting the values corresponding to the longest treatment duration with the tau lentiviruses to 100% per experiment and re-scaling the resting values accordingly. At least two replicate wells were included per condition. Data points are single-well observations representing the average of 40 fields containing information from >1000 neurons. The total number of neurons analyzed per group is indicated in Appendix A.

### 4.7. Transmission Electron Microscopy (TEM) Analysis

For TEM, neurons were fixed on DIV 18 with 2.5% glutaraldehyde (Merck, Rahway, NJ, USA) in 0.1 M cacodylate buffer, pH 7.4. After fixation, cells were washed three times for 5 min with 0.1 M cacodylate buffer, pH 7.4, and postfixed for 1 h at RT with 1% OsO_4_ (Aurion)/1% KRu(CN)_6_ (Sigma-Aldrich). After dehydration through a series of increasing ethanol concentrations, cells were embedded in Epon and polymerized for 48 h at 60 °C. The coverslip was removed by alternately dipping it in hot water and liquid nitrogen. The flat Epon-embedded cell monolayer was mounted on pre-polymerized Epon blocks for thin sectioning. Ultrathin sections (80 nm) were cut parallel to the cell monolayer, collected on formvar-coated single-slot copper grids, and stained with 0.5% uranyl acetate and Reynolds lead citrate in an ultrastainer (EM-AC20, Leica, Wetzlar, Germany). Filamentous structures were imaged on a Tecnai12 biotwin transmission electron microscope with a side-mounted CCD camera (Velata, Olympus Soft Imaging Solutions) at 20,500 and 43,000 times magnification. The analysis of filamentous structures was performed with ImageJ2 software (v2.9.0/1.53t; National Institutes of Health). The diameter of the filaments was measured by perpendicular line measurements. The organization of filaments in the aggregate was quantified by measuring the angle and length of individual filaments in the ultrathin cross section. To assess to what extent aggregates contain parallel-stacked filaments, we calculated the SD of the filament angle within aggregates. Measurements were performed for hollow filaments (putative microtubules) present in the same micrographs as the solid tau filaments as a reference. The number of analyzed filaments per group can be found in Appendix A.

### 4.8. Statistics

Statistical analysis and plotting of data were performed using Graphpad Prism version 8.4.2 (Graphpad) software. The normality of the data distribution was assessed with the Shapiro–Wilk test. The analysis of normally distributed data was done with nested analyses, which fit a mixed-effects model to account for dependency among data points obtained from the same experiment, i.e., neurons obtained from mice in the same litter. A nested t-test or nested one-way ANOVA with Tukey’s post-hoc test were employed when comparing two or more groups, respectively. In cases where one or more datasets were not normally distributed, Kruskal–Wallis with Dunn’s post-hoc test was used. The statistical tests used, the number of independent experiments (corresponding to the number of pregnant mice), the detailed and statistical results per experiment are indicated in Appendix A. Statistical significance was considered when the *p* value < 0.05. *p* values are indicated in the figure graphs as: * *p* < 0.05, ** *p* < 0.01, *** *p* < 0.001, **** *p* < 0.0001, n.s. not significant.

## Figures and Tables

**Figure 1 ijms-24-10865-f001:**
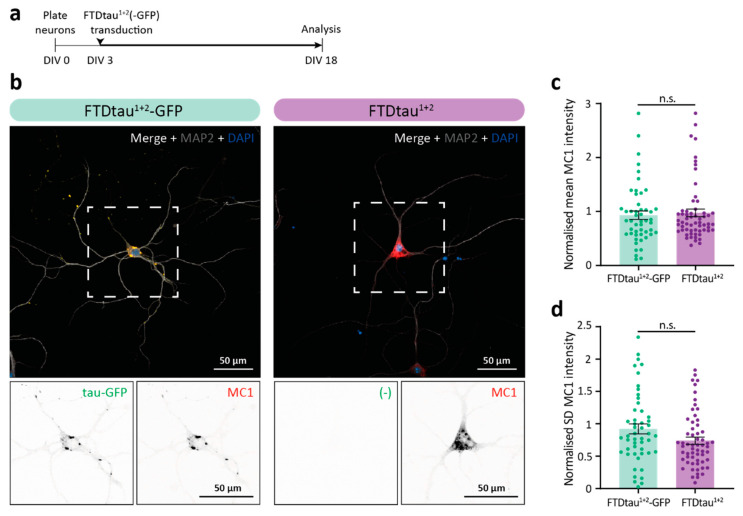
Pathological FTDtau^1+2^-GFP and FTDtau^1+2^ assemblies have different morphology and subcellular localisation. (**a**) Schematic representation of the protocol to induce seed-independent intraneuronal tau pathology in primary mouse neurons. Wild type (WT) neurons were plated at DIV 0. Transduction with FTDtau^1+2^-GFP or FTDtau^1+2^ was at DIV 3, and samples were fixed and analyzed at DIV 18. (**b**) Representative confocal images of neurons transduced with FTDtau^1+2^-GFP and FTDtau^1+2^. Tau-GFP signal is shown in green, and nuclei are visualized in blue by 4′,6-diamidino-2-phenylindole (DAPI). Immunofluorescence was performed for the neuron-specific dendritic marker microtubule associated protein-2 (MAP2; shown in gray) and MC1 as a marker for pathological conformations of tau (shown in red). Separate channels are shown in grayscale. (**c**,**d**) Single-cell quantification of tau pathology load in neuronal somata, represented as mean (**c**) and standard deviation (**d**) values. Data points represent single neuron values, and error bars represent the standard error of the mean (SEM). N = 3 independent experiments. n.s. = not significant, nested t-test.

**Figure 2 ijms-24-10865-f002:**
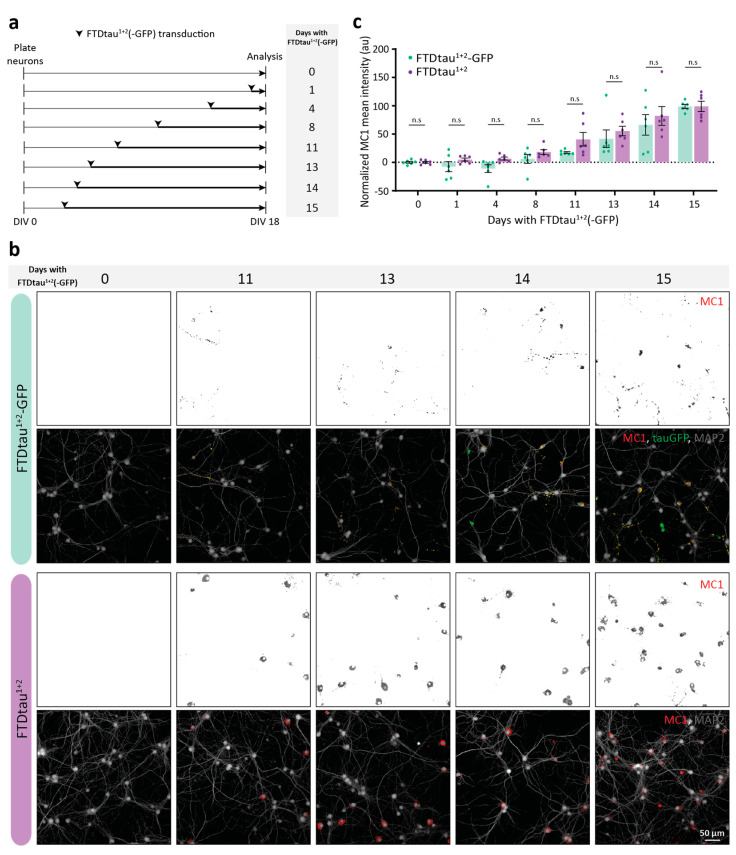
FTDtau^1+2^-GFP and FTDtau^1+2^ pathology progresses with similar kinetics. (**a**) Schematic representation of the experimental set-up to assess the progression of tau pathology in the FTDtau^1+2^-GFP and FTDtau^1+2^ models. Experiment duration (plating and fixation days) was the same as the regular protocol (Figure 1a), whereas the time point for lentiviral transduction changed, resulting in different exposure durations per condition. (**b**) Representative epifluorescence images for selected time points. Immunofluorescence was performed for the neuron-specific dendritic markers MAP2 (grey) and MC1 (red). In FTDtau^1+2^-GFP condition, tau’s fluorescent signal is shown in green. Separate channels are shown in grayscale. (**c**) Quantitative high-content automated microscopy analysis of the mean MC1 immunofluorescence intensity for FTDtau^1+2^-GFP and FTDtau^1+2^ models per time point, relative to 15 days of exposure (100%). N = 3 independent experiments, n = 6 observations per time point. Data points represent the corrected mean value per well. Error bars represent the SEM. n.s. = not significant, nested t-test for comparison between models per time point.

**Figure 3 ijms-24-10865-f003:**
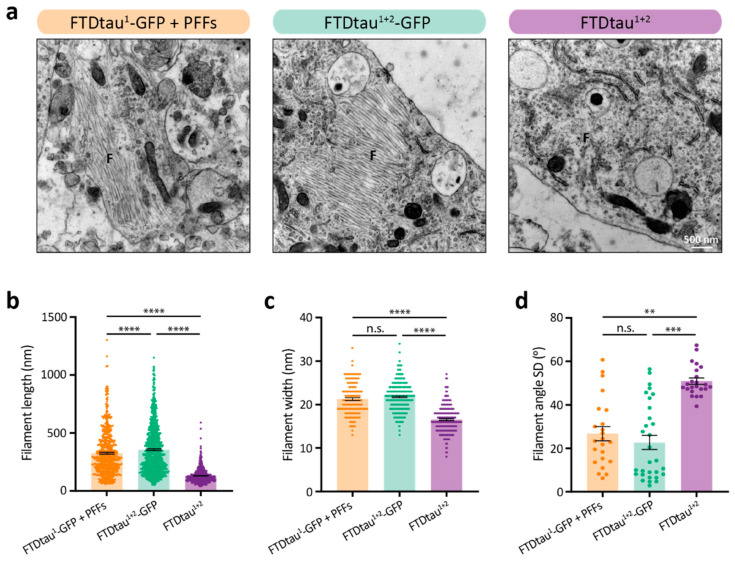
Presence of GFP tag alters the ultrastructure of FTDtau^1+2^ filaments. (**a**) Representative TEM images of neuronal somata showing fibrillar structures (F) in the different tau pathology models studied. (**b**–**d**) Quantitative analyses of filament properties consisting of filament length (**b**), filament width (**c**), and SD of filament angle as bundling parameter (**d**). Data points in (**b**,**c**) represent single filament values, while data points in (**d**) correspond to the SD of filament angle per analyzed micrograph. Error bars represent the SEM. N = 3 independent experiments. ** *p* < 0.01; *** *p* < 0.001; **** *p* < 0.0001; n.s. = not significant, Kruskal-Wallis with Dunn’s post-hoc test for not normally distributed data.

**Figure 4 ijms-24-10865-f004:**
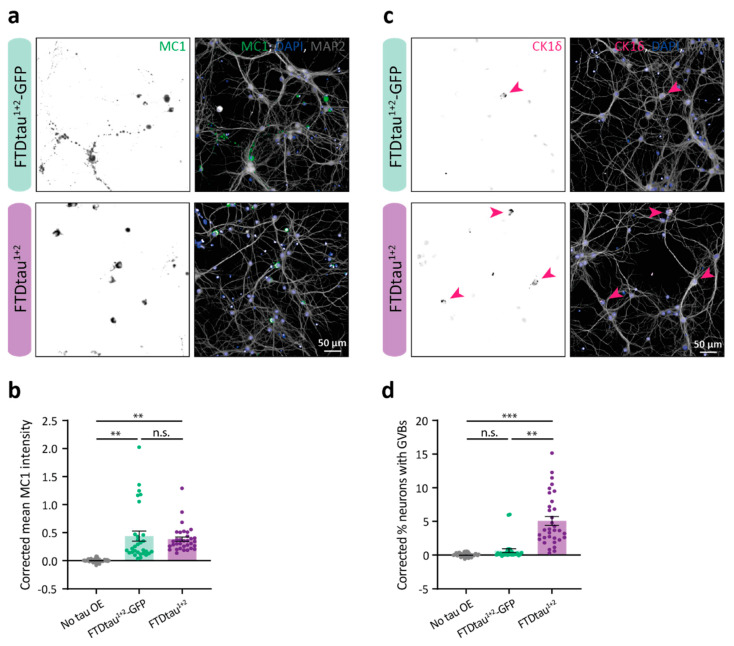
FTDtau^1+2^ is associated with a higher GVB load than FTDtau^1+2^-GFP. (**a**,**c**) Representative epifluorescence images for MC1-positive tau accumulation (**a**) and GVBs (**c**) in primary neuron cultures transduced with either FTDtau^1+2^-GFP and FTDtau^1+2^ following the general protocol (Figure 1a). Arrowheads point at neurons with CK1δ-positive GVBs. Separate channels are shown in grayscale. (**b**,**d**) Quantitative high-content automated microscopy analysis of tau pathology parameters represented in (**a**,**c**). A condition that was not transduced with a tau-overexpressing virus was included as control. N = 8 independent experiments. Data points represent corrected values per well, and error bars represent the SEM. (**b**) Mean MC1 immunofluorescence intensity. ** *p* < 0.01; n.s. = not significant, Kruskal–Wallis with Dunn’s post-hoc test. (**d**) Percentage of neurons with GVBs. ** *p* < 0.01; *** *p* < 0.001; n.s. = not significant, nested one-way ANOVA followed by Tukey’s post-hoc test.

**Figure 5 ijms-24-10865-f005:**
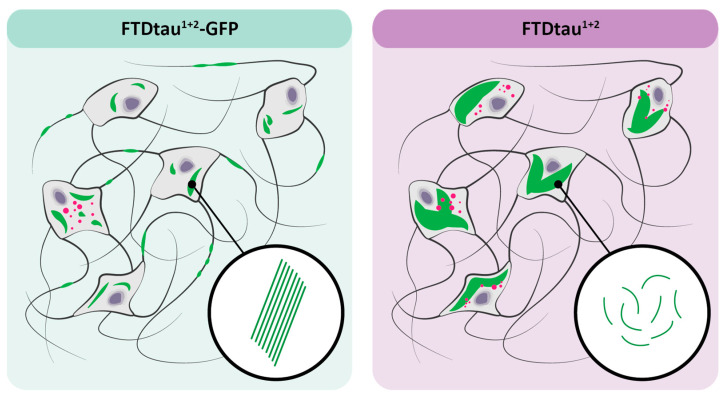
FTDtau^1+2^-GFP and FTDtau^1+2^ tau form structurally and morphologically distinct pathological tau assemblies that differentially affect GVB accumulation. Schematic overview of the results: The FTDtau^1+2^-GFP tau pathology model results in densely packed, parallel tau filaments (in green), which are present in both the neuronal soma and neurites. Tau pathology in the FTDtau^1+2^ model is more evenly distributed within neuronal somata and presents a less organized filament ultrastructure. In addition, the FTDtau^1+2^ filaments are thinner and shorter than the FTDtau^1+2^-GFP filaments. The GVB load (in magenta) in the FTDtau^1+2^ model is approximately 10-fold higher than in the FTDtau^1+2^-GFP model.

## Data Availability

Data are available upon reasonable request to the corresponding author.

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
