# Peer review of "Structurally and Morphologically Distinct Pathological Tau Assemblies Differentially Affect GVB Accumulation"

_ijms, 2023, doi:10.3390/ijms241310865_

Round 1

Reviewer 1 Report

Interesting study looking at effects of GFP tagged Tau compared to untagged. Studying the differences that tau aggregation inhibitors will have on the on GFP tagged vs untagged Tau could provide further insights.

Author Response

We thank the reviewer for their positive evaluation of the manuscript. In the discussion section, we have now addressed the interesting suggestion of the reviewer to employ aggregation inhibitors in the different tau aggregation models to obtain further mechanistic insight (lines 442-444). 

Reviewer 2 Report

The manuscript entitled “Structurally and morphologically distinct pathological tau assemblies differentially affect GVB accumulation” by Jorge-Oliva et al. compared the pathological tau assemblies induced by FTDtau1+2 or FTDtau1+2 with a GFP tag at the C-terminus. Both approaches caused spontaneous pathological tau accumulation in the cultured neurons but with different vesicular morphologies as well as distinct GVB accumulation. In sum, GFP expression could affect tau accumulation and reduce GVB formation. The study convincingly showed spontaneous tau pathology through immunofluorescence staining and transmission electron microscopy. Interestingly, the co-expression of another irrelevant protein (GFP) changed the distribution of the tau assemblies and the formation of the GVBs. The data can support the conclusions and I have several questions.

1. The supplementary data are missing from the submitted manuscript instead it seems the authors mistakenly uploaded a published paper from the same group.

2. The authors claim that in the FTDtau1+2-GFP model, the subcellular localization and morphology of MC1-positive tau assemblies resembles the aggregation of tau from the FTDtau1-GFP+K18 P301L tau PFFs model. So I wondered if the tau aggregations from FTDtau1+2 (no GFP) look similar to those from FTDtau1 (no GFP)+K18 P301L tau PFFs?

3. Since FTDtau1+2 displayed more GVB than FTDtau1+2-GFP, it will be interesting to understand the mechanisms of how it happens. Would it be possible that GFP protein consumes the vesicles which prevent it from forming the GVB? Can the authors discuss more about it?

4. Have the authors tried to check if the tau assemblies and GVB will form under the condition of FTDtau1 in the presence of amyloid fibrils?

Author Response

We thank the reviewer for their positive assessment of our manuscript and helpful suggestions which we used to improve the manuscript as detailed point-to-point below.

  1. We apologize for our mistake in uploading the wrong file. The correct file with the supplementary data has now been uploaded with the revision. 
  2. Our data indicate that the GFP tag and not the FTD mutation(s) or the requirement for seeding with PFFs is the key determinant in the different subcellular localization and morphology of MC1-positive tau assemblies. In line with this it would indeed be expected  the assemblies found with untagged FTDtau1+2 (no GFP) look similar to those from untagged FTDtau1 +K18 P301L tau PFFs. We elaborated on this in the discussion (lines 410-413). 
  3. We have included the interesting explanation of the reviewer (sequestration of GVB forming vesicles by GFP) as potential mechanism for the difference in GVB formation between the GFP-tagged and untagged tau aggregation models in the discussion (lines 454-461).
  4. We thank the reviewer for this interesting idea. Although this is beyond the scope of the current manuscript, we have further elaborated on the potential implications of our findings in the context of beta-amyloid pathology as present in the AD brain in the discussion section (lines 451-453). 

Reviewer 3 Report

Unable to access the results data from supplementary table as author erroneously enclosed other manuscript in place of supplementary data. Please rectify it and share the correct information for further review.  

No

Author Response

We apologize for our mistake in uploading the wrong file. The correct file with the supplementary data has now been uploaded with the revision. 

Reviewer 4 Report

The manuscript entitled "Structurally and morphologically distinct pathological tau assemblies differentially affect GVB accumulation" by Oliva et al. is well-designed and executed by the authors.

I have a minor suggestion to improve the manuscript.

1. Summarize these findings as a figure.

2. Add the micromolar uniformly in the entire manuscript. 

Author Response

We thank the reviewer for their positive assessment of our manuscript and helpful suggestion that we have used to improve the manuscript as detailed point-to-point below:

1. Following the reviewer's suggestion we have now summarized our findings in an additional figure (Figure 5; lines 394-402).

2. We thank the reviewer for pointing this out. We checked the manuscript and found one instance where um and not µM was used, it has been corrected (line 134).